# Interleukin-1α Is a Critical Mediator of the Response of Human Bronchial Fibroblasts to Eosinophilic Inflammation

**DOI:** 10.3390/cells10030528

**Published:** 2021-03-02

**Authors:** Ksenija Bernau, Jonathan P. Leet, Heather Floerke, Ellen M. Bruhn, Andrea L. Noll, Ivy S. McDermott, Stephane Esnault, Nizar N. Jarjour, Nathan Sandbo

**Affiliations:** Department of Medicine, Division of Allergy, Pulmonary and Critical Care Medicine, School of Medicine and Public Health, University of Wisconsin-Madison, Madison, WI 53792, USA; jleet@wisc.edu (J.P.L.); hlfloerke@medicine.wisc.edu (H.F.); ebruhn@dermatology.wisc.edu (E.M.B.); alnoll@wisc.edu (A.L.N.); imcdermott@wisc.edu (I.S.M.); sesnault@medicine.wisc.edu (S.E.); nnj@medicine.wisc.edu (N.N.J.); nsandbo@medicine.wisc.edu (N.S.)

**Keywords:** eosinophil, fibroblast, asthma, IL6, IL8, IL1α, IL1 receptor, cytokines, intracellular signaling

## Abstract

Eosinophils contribute to allergic inflammation in asthma in part via elaboration of a complex milieu of soluble mediators. Human bronchial fibroblasts (HBF) respond to stimulation by these mediators by acquiring a pro-inflammatory profile including induction of interleukin 6 (IL6) and IL8. This study sought to determine key component(s) of eosinophil soluble factors that mediate IL6 and IL8 induction in HBF. HBF treated with eosinophil-derived soluble mediators were analyzed for gene expression, intracellular signaling, and IL6 and IL8 secretion following inhibition of inflammatory signaling. Segmental allergen bronchoprovocation (SBP-Ag) was performed in mild asthmatics and bronchoalveolar lavage fluid was analyzed for eosinophils and cytokines. We found that signaling via the IL1α/IL1 receptor is an essential component of the response of HBF to eosinophil-derived soluble factors. IL1α-dependent activation of nuclear factor kappa-light-chain-enhancer of activated B cells (NFκB) signaling is required to induce IL6 secretion. However, NFκB signaling is dispensable for the induction of IL8, whereas Src is required. IL1α is associated with eosinophilic inflammation in human airways after SBP-Ag. Conclusions: IL1α appears to be a critical component of the soluble eosinophil-derived milieu that drives pro-inflammatory bronchial fibroblast responses and associates with eosinophilic inflammation following SBP-Ag. Disruption of IL1α-signaling could modify the downstream effects of eosinophilic inflammation on airway remodeling.

## 1. Introduction

Eosinophilic inflammation is a hallmark of allergic asthma. Eosinophil presence in the asthmatic airway is associated with increased disease severity and risk of exacerbations [1,2]. Airway eosinophils contribute to allergic inflammation in asthma by releasing a variety of soluble mediators and cell-free granule proteins through cytolysis and degranulation [3,4,5]. These factors contribute to the pro-inflammatory activation of other cell types, including bronchial fibroblasts, resulting in prolonged tissue injury that can lead to airway remodeling and subsequent irreversible airflow obstruction in some subjects [6,7,8]. The presence of irreversible airflow obstruction poses a major treatment dilemma in some severe asthmatics, given the paucity of effective therapeutics that target airway remodeling.

Eosinophilic inflammation plays an essential role in airway remodeling [9], and while the role of the bronchial fibroblast in asthma tissue remodeling is equally evident [10,11], a detailed understanding of the interplay between airway eosinophilia and bronchial fibroblast activation is lacking, holding back the development of targeted therapeutics to address this problem. In asthmatic airways, activated eosinophils release a myriad of soluble and insoluble components (see Review [12]), including granules containing major basic protein, eosinophil cationic protein (ECP), and eosinophil-derived neurotoxin (EDN), which can cause direct damage to airway epithelium [13]. Eosinophils have been observed to release cytokines important in fibroblast activation and function, including transforming growth factor β (TGF-β) [14], and co-culture of eosinophils with fibroblasts promotes the production of extracellular matrix (ECM) components [15,16]. However, previous studies have typically utilized approaches where identified eosinophil-derived mediators are tested in isolation to observe effects on secondary structural cell cultures. While these studies have led to a critical mechanistic understanding of potential signaling that may result from these mediators, it is unclear how the milieu of the released products by eosinophils affects bronchial fibroblasts in aggregate.

In our previous work, we sought to begin to answer this question by utilizing an ex vivo model of human primary eosinophil cultures that both recapitulates the phenotype of asthmatic airway eosinophils and also enables inducible cytolysis and degranulation on heat-aggregated immunoglobulin G (IgG), mimicking eosinophil interactions with immunoglobulins found in vivo [5,6,17]. Using this model, we isolated the supernatants from cytolyzing and degranulating eosinophils, and, using a global transcriptomic approach, found that exposure of human lung fibroblasts to the complete milieu of eosinophil-derived soluble mediators results in the development of a pro-inflammatory gene expression profile [7,17]. We additionally found that stimulation of human bronchial fibroblasts (HBF) with eosinophil-derived soluble mediators results in the elaboration of two key inflammatory factors, interleukin 8 (IL8) and interleukin 6 (IL6) [6]. However, the dominant component(s) released from the degranulating/cytolyzing eosinophils that drive these airway fibroblast responses is not known. Because identification of these components would lend important insight into how bronchial fibroblasts respond in aggregate to eosinophilic inflammation and provide a potential target for intervention, in this study, we sought to characterize the primary activator and signals driving bronchial fibroblast responses to eosinophil-derived soluble mediators. Using our previous transcriptomic data and pathway analysis as a guide [7], we hypothesized that eosinophil soluble mediators signal to HBF through the IL1 receptor (IL1R) leading to the pro-inflammatory fibroblast phenotype. In this study, we have found that upon stimulation with eosinophil soluble mediators, HBF are activated through IL1R by IL1α, but not IL1β, leading to the activation of divergent intracellular signaling pathways that lead to the release of IL6 and IL8.

## 2. Materials and Methods

### 2.1. Human Subjects and Cell Preparation

This study was conducted according to the guidelines of the Declaration of Helsinki and approved by the University of Wisconsin-Madison Health Sciences Institutional Review Board (IRB#2013-1570, IRB#1999-292). All subjects provided written consent prior to participation.

Circulating blood eosinophils were obtained from donors with known allergy, rhinitis and/or asthma who underwent phlebotomy. Eosinophils were purified by negative selection as previously described [17]. Briefly, heparinized blood was diluted in Hank’s balanced solution (HBSS, 1:1), overlaid above Percoll (1.090 g/mL), centrifuged at 700× *g* for 20 min at room temperature (RT). Mononuclear cells were subsequently removed from the interface between Percoll and plasma, while red blood cells were lysed from the pellet. The remaining pellet containing polymorphonuclear leukocytes was suspended in 2% calf serum in HBSS and then incubated with anti-CD16, anti-CD3, anti-CD14, and anti-glycophorinA immunogenic beads (Mylteni, Bergisch Gladbach, Germany), leaving behind eosinophils with a purity and survival of >99%.

HBF were obtained from healthy non-atopic, non-smoking adult donors using bronchoscopy-driven bronchial biopsy specimens that were then de-identified. Bronchoscopy specimens were histologically assessed to confirm normal bronchial tissue architecture, and HBF were derived as before [18]. Briefly, tissue pieces were digested in fibroblast starvation medium (FGM) Bulletkit medium (CC-33132 Lonza, Basel, Switzerland) with collagenase H (1 mg/mL) at 4 °C before being cultured in fibroblast growth medium, which included: FGM Bulletkit medium, 2% fetal bovine serum (FBS), human recombinant insulin (CC-4021J, Lonza), recombinant human fibroblast growth factor-B (CC-4065J, Lonza) and gentamycin sulfate amphotericin B (GA1000, CC4081J, Lonza). A homogenous fibroblast population was established by expanding fibroblasts every several days.

We recruited 18 subjects who had a history of mild asthma with airway reversibility to albuterol and a positive skin prick test to one or more aeroallergens, to undergo in vivo segmental bronchoprovocation with antigen (SBP-Ag). The subjects were nonsmokers and did not have a respiratory infection or asthma exacerbation within 30 days of study, and had not received long-acting β-agonists within two days, antihistamines or leukotriene antagonists within seven days, or corticosteroids within 30 days of study enrollment. Bronchoscopy, bronchoalveolar lavage (BAL) and SBP-Ag were performed as previously described [19,20]. Briefly, the antigen dose leading to 20% forced expiratory volume in 1 sec (FEV_1_) fall (Ag PD_20_) was calculated from a dose–response curve generated by a graded inhaled antigen challenge. A total dose of 30% of the antigen PD_20_ was administered for SBP-Ag; 10% in one segment and 20% in a second segment. In all subjects, BAL was performed in each segment before and 48 h after SBP-Ag. BAL fluid from the two segments was pooled for fluid and cell analysis. Cell differentials were determined after cytospin and staining with Wright-Giemsa-based Hema-3 while BAL fluids were examined via ELISA (described below).

### 2.2. Cell Cultures

Eosinophils were cultured at 1 × 10^6^ cells/mL in medium containing RPMI 1640 with L-glutamine and 25 mM HEPES (Corning, Corning, NY, USA), 10% FBS (Gibco, Thermo Fisher Scientific, Verona, WI, USA) with antibiotic/antimyotic (Gibco), 2 mM L-glutamine (Gibco) and 100 mg/mL ciprofloxacin-HCL, and IL3 (4 ng/mL, R&D Systems Inc., Minneapolis, MN, USA) for 20 h. Concurrently, heat aggregated human IgG (IgG) was prepared for 30 min at 63 °C in phosphate-buffered saline (PBS), as previously described [17]. After 20 h of incubation with IL3, eosinophils were washed and suspended at 1 × 10^6^ cells/mL in new medium without IL3, and 1 × 10^6^ cells were moved to a 24-well plate that had been previously coated overnight with IgG (10 µg/mL; 500 µL/well, I-2511 Sigma Aldrich, St. Louis, MO, USA) and saturated with 0.1% gelatin for 30 min at 37 °C in PBS. After 6 h of incubation on IgG, eosinophil supernatant fluids were collected and stored at −80 °C for subsequent use for activation of HBF.

HBF were maintained on tissue culture plastic and utilized for experiments between Passages 2 and 7. For experiments, HBF were plated at a density of 150,000 cells/mL in fibroblast growth medium for 24 h prior to serum starving in fibroblast starvation medium (FGM Bulletkit medium with gentamycin sulfate amphotericin B and 0.4% FBS) for 24 h. HBF inhibitors, including IL1 receptor antagonist (IL1RA, 100 ng/mL, R&D Systems, Inc.), inhibitor of Src-family kinases (PP2 and control (PP3), 10 µM, Cayman Chemical, Ann Arbor, MI, USA), IκB kinase inhibitor (BMS-345541, 10 µM, Cayman Chemical), and Janus-associated kinase (JAK) inhibitor (Ruxolitinib, 100 nM, Cayman Chemical), were added to HBF medium for 30 min prior to stimulation with eosinophil supernatant fluids or control medium (1:1 ratio with fibroblast starvation medium). Conversely, neutralizing antibodies, including anti-IL1α (1 µg/mL, mabg-hil1a-3, InvivoGen, San Diego, CA, USA) and anti-IL1β (1 µg/mL, mabg-hil1b-3, InvivoGen), as well as IgG1 isotype control (1 µg/mL, mabg1-ctrlm, InvivoGen), were incubated with eosinophil supernatant fluids for 1 h at 37 °C prior to being added to HBF. HBF were treated with recombinant human IL1α (1 ng/mL, 200-LA, R&D systems) or IL1β (1 ng/mL, 201-LB, R&D systems) agonists or eosinophil supernatants with or without inhibitors for 30 min or 24 h, at which point HBF supernatants were collected (24 h incubation) while the cells were lysed for RNA (24 h incubation) or protein (30 min incubation) isolation.

### 2.3. ELISA

Detection of IL6 and IL8 was performed by enzyme-linked immunosorbent assay (ELISA), as previously described [21], using the sandwich method. ELISA plate coating was done using mouse monoclonal anti-human IL8 (clone G265-5, BD Biosciences, San Jose, CA, USA) and IL6 (clone 6708 R&D Systems, Inc.). For detection, biotinylated mouse monoclonal anti-human IL8 (clone G265-8, BD Biosciences) and biotinylated polyclonal goat anti-human IL6 (R&D Systems, Inc.) were utilized with sensitivities of less than 3 pg/mL for IL8 and IL6.

BAL fluids were retrieved before and 48 h after SBP-Ag, concentrated 20-fold and IL1α and IL1β concentrations were determined using a DuoSet Development kit (DY-200 and DY-201, respectively, R&D Systems, Inc.), while EDN concentrations were determined using the sandwich ELISA kit (7630, MBL International Corp., Des Plaines, IL, USA). The IL1α and IL1β DuoSet Development ELISA kits and EDN ELISA kit had sensitivities of 1 pg/mL and 1 ng/mL, respectively.

### 2.4. Reverse Transcription Quantitative Real-Time PCR

Reverse transcription quantitative real-time polymerase chain reaction (RT-qPCR) was performed as before [22]. Briefly, pre-treated HBF RNA was harvested using RNA STAT-60 (AMS Biotechnology, Milton, Abingdon, UK). Approximately 1 µg of total HBF RNA was utilized for random-primed reverse transcription using an iScript cDNA synthesis kit (Bio-Rad, Hercules, CA, USA). RT-qPCR was done using iTaq SYBR Green supermix with ROX (Bio-Rad) in an Applied Biosystems 7500 multicolor real-time PCR detection system (Applied Biosystems). The following primer pair sequences were used with glucuronidase β (GUSB) serving as a housekeeping gene: GUSB forward (CAGGACCTGCGCACAAGAG), GUSB reverse (AGCGTGTCGACCCCATTC), CXCL8 forward (CTTGGCAGCCTTCCTGATTT), CXCL8 reverse (TTCTTTAGCACTCCTTGGCAAAA), IL6 forward (TGCAGATGAGTACAAAAGTCCTGAT), IL6 reverse (GTGGTTATTGCATCTAGATTCTTTGC). Prior to experimental RT-qPCR, standard curves and primer efficiencies were determined, resulting in the following efficiencies: 93% (GUSB), 101% (CXCL8), 95% (IL6).

### 2.5. Western Blot

Following 30 min of treatment with eosinophil supernatants with or without inhibitors, HBF were lysed for protein and subsequently subjected to sodium dodecyl sulphate–polyacrylamide gel electrophoresis (SDS-PAGE) and Western blotting, as before [23]. Briefly, cell lysis was done in radioimmune precipitation assay (RIPA), containing 25 mM HEPES (pH 7.5), 150 mM NaCl, 1% Triton X-100, 0.1% SDS, 2 mM ethylenediaminetetraacetic acid (EDTA), 2 mM egtazic acid (EGTA), 10% glycerol, 1 mM NaF, 200 µM sodium orthovanadate, and protease inhibitor mixture (Sigma Aldrich, St. Louis, MO, USA), for 10 min on ice, followed by centrifugation at 21.1× *g* for 10 min at 4 °C. Supernatants were mixed with Laemmli buffer, boiled for 5 min before being subject to SDS-PAGE and Western blotting against indicated primary and appropriate HRP-conjugated secondary antibodies. Rabbit polyclonal antibody was against pSTAT3 (ab76315 Abcam, Cambridge, UK), while mouse monoclonal antibodies were against NFκB inhibitor α (IkBα) (MAB4299, R&D Systems), IkBα (ab12134, Abcam), phospho-IkBα (MA5-15224, Invitrogen, Carlsbad, CA, USA), and β-actin (A1978, Sigma Aldrich). Enhanced chemiluminescence (ECL) reaction reagents and GE LAS4000 were used to visualize immunoreactive bands (GE Healthcare, Little Chalfont, UK). Western blot images were linearly adjusted to aid in the visualization of the relevant band. Unadjusted bands were quantified using ImageJ 1.48v [24].

### 2.6. Statistical Analyses

Densitometric Western blot and RT-qPCR data were assessed by representing data as fold of control and utilizing Student’s unpaired *t*-test with Bonferroni correction where applicable. Levels of IL6 and IL8 measured by ELISA were compared by log10 transforming the data and analyzing using Student’s paired *t*-test. Change in EOS%, EDN, IL1α or IL1β concentration after (D2) minus before (D0) SBP-Ag measured by ELISA were calculated and a Spearman correlation was used for statistical analyses. Note that EOS%, EDN, IL1α and IL1β levels before challenge are typically below 1% and below detection, respectively.

## 3. Results

### 3.1. IL6 and IL8 Secretion from HBF Treated with Eosinophil-Derived Supernatants Is Dependent on the IL1 Receptor

As a first approach, we sought to explore the potential role of IL1 family cytokines in mediating fibroblast responses from factors released by eosinophils. We have previously identified the release of IL1β from activated eosinophils [25,26], which could account for paracrine signaling in the asthmatic airway. In addition, previous transcriptomic data of the response of lung fibroblasts to eosinophil-derived soluble mediators (using the same methodology) followed by pathway analysis suggested that IL1 signaling was a potential upstream mediator of fibroblast responses [7]. To determine if IL1 drives fibroblast responses to eosinophil-derived soluble mediators, we first utilized the IL1 receptor antagonist (IL1RA), the competitive endogenous protein inhibitor of the IL1R. As shown in Figure 1A,B and as previously reported, there is marked induction of IL8 and IL6 secretion by airway fibroblasts in response to treatment with supernatants from degranulating eosinophils, as determined by ELISA. Of note, we have previously demonstrated that eosinophil supernatants do not contain elevated amounts of IL6 or IL8 [6]. HBF treatment with IL1RA 30 min prior to stimulation with eosinophil-derived supernatants results in a marked attenuation of IL8 and IL6 protein release (Figure 1A,B), suggesting that either expression or secretion of these two cytokines is dependent on IL1 signaling. We then examined the effect of IL1RA inhibition on HBF-induced CXCL8 and IL6 transcription by eosinophil-derived products. We observed a marked attenuation in the mRNA transcripts for both genes (Figure 1C,D), indicating that signaling via IL1R is essential for the transcriptional upregulation of these two genes in response to eosinophil-derived mediators.

### 3.2. IL1β Is Dispensable for the Induction of IL6/IL8 in HBF Treated with Eosinophil-Derived Supernatants

Because IL1β can be released by activated eosinophils [25,26], we tested whether IL1β may be the primary agonist of the IL1 receptor in this context. As shown in Figure A1(1,2), the maximal response of the HBF release of IL6 or IL8 to recombinant (r)IL1β stimulation occurs at 1 ng/mL (55 pM). To do so, we first validated the inhibitory capacity and specificity of the neutralizing anti-IL1β antibody (clone 4H5, InvivoGen) from 10 ng/mL (68 pM, ~equimolar) to 10 µg/mL (~1000-fold molar excess of anti-IL1β neutralizing antibody). Incubation of the anti-IL1β neutralizing antibody (or isotype control IgG) with 1 ng/mL rIL1β for 1 h prior to stimulation results in a dose-dependent blockade of IL6 and IL8 release by HBF compared to IgG control antibodies (Figure A2(1,2)), suggesting that this antibody is effective in blocking IL1β effects. Likewise, this antibody does not block rIL1α-induced effects, confirming the specificity of the blockade (figure not shown). We then utilized the antibody, at a previously optimized concentration of 1 µg/mL, to determine the role of IL1β in eosinophil supernatant-mediated stimulation of airway fibroblasts. As shown in Figure 2, incubation of the IL1β antibody with eosinophil supernatants for 1 h prior to stimulation does not attenuate IL8 or IL6 secretion by airway fibroblasts.

### 3.3. IL1α Is Essential for the Induction of IL6/IL8 in HBF Treated with Eosinophil-Derived Supernatants

Considering that the blockade of IL1R, but not the neutralization of IL1β agonist, results in significant inhibition of IL6 and IL8 release, we sought to examine whether IL1α was the primary agonist signaling via the IL1R in HBF. As with IL1β, we first established the optimal concentration of the rIL1α agonist to be 1 ng/mL (Figure A1(3,4)) and then validated the anti-IL1α antibody (clone 7D4), observing that it is capable of blocking the rIL1α-dependent release of IL6 and IL8 by HBF in a dose-dependent manner (Figure A2(3,4)). Similar to IL1β, we chose 1 µg/mL (6.8 nM, 124-fold molar excess) as the optimized blocking dose. Additionally, IL1α neutralizing antibody has no effect on rIL1β-induced IL6 and IL8 expression (Figure A3). Subsequently, we employed 1 µg/mL of IL1α neutralizing antibody to examine its ability to disrupt HBF pro-inflammatory activation. We found that treatment of IL3IgG-activated eosinophil supernatants with the anti-IL1α neutralizing antibody for 1 h prior to incubation with HBF markedly reduces the ability of these cells to release both IL6 and IL8 (Figure 3). These data suggest that eosinophil lysis products lead to a release of IL6 and IL8 through activation by IL1α.

### 3.4. IL1 (α/β) Is Associated with Eosinophilic Inflammation after SBP-Ag

In light of these observations indicating that IL1α is playing a role in the activation of HBF to release pro-inflammatory mediators in vitro, we were interested in examining the relationship between IL1α and eosinophil activation in an in vivo model of allergic inflammation, such as that seen in asthma. For this, we utilized BAL samples collected from patients with mild allergic asthma who underwent SBP-Ag (see supplemental Table A1 in Appendix D and [21] for patient characterization). In this human model of allergic inflammation, BAL eosinophils represent less than 1% of total BAL cells prior to SBP-Ag (baseline, D0), but markedly increase 48 h later [19,27]. In these samples, we now find that the levels of IL1α are undetectable in most patients (17 out of 18) on D0. Forty-eight h after SBP-Ag challenge, the levels of IL1α markedly increase in the BAL fluids of subjects with the most response to the allergen (as determined by the eosinophil %), demonstrating a correlation between the presence of IL1α and % eosinophils detected in the BAL (Figure 4A). We also observed a similar correlation between IL1β and BAL eosinophils % (Figure 4B). Furthermore, when we assayed the amount of eosinophil-derived neurotoxin (EDN), a marker of eosinophil degranulation, we found correlation with IL1α and IL1β levels (Figure 4C,D). This suggests that increases in eosinophilic inflammation and degranulation are associated with increases in IL1α, providing an in vivo relevant link to the IL1α-dependent effects that we observed in fibroblasts in vitro. Changes in % of the other cell types counted in the BAL fluids (macrophages, neutrophils and lymphocytes; supplemental Table A2 in Appendix E) did not display correlation with either IL1α or IL1β (not shown).

### 3.5. Release of IL6 in HBF Stimulated with Eosinophil-Derived Supernatants Is Dependent on NFkB Signaling

Because IL1α signals canonically via the IL1R through IL1 receptor-associated kinase 1 (IRAK1), MyD88 and nuclear factor kappa-light-chain-enhancer of activated B cells (NFκB) signaling, we then sought to examine the activation of this pathway by eosinophil-derived supernatants. As show in Figure 5, the treatment of airway fibroblasts with eosinophil-derived supernatants induces IκBα phosphorylation at Ser 32/36 residues and the degradation of IκBα, indicating activation of IκB kinase (Ikk) and release of Rel65 and Rel50 (NFκB) for nuclear translocation. This indicates that, at 30 min, there is activation of NFκB signaling in HBF by eosinophil-derived supernatants. When eosinophil supernatants were incubated with IL1α (but not IL1β) neutralizing antibody prior to stimulation, IκBα phosphorylation was inhibited in HBF (Figure 5A,B), suggesting that the activation of NFκB by eosinophil supernatants is dependent on IL1α. Consistent with this finding, the blockade of NFκB signaling by the IκB kinase inhibitor, BMS-345541, strongly attenuated IL6 secretion by HBF in response to eosinophil-derived supernatants (Figure 5C), establishing the requirement for NFκB signaling in IL6 induction.

### 3.6. IL6 Expression by HBF Requires Janus Kinase (JAK)/Signal Transducer and Activator of Transcription Protein (STAT) Signaling

We also observed the induction of STAT3 by eosinophil supernatants (Figure 6A). However, incubation with IL1α and IL1β neutralizing antibodies did not affect phosphorylation of STAT3 (Figure 6A,B). Nevertheless, inhibition of eosinophil supernatant-induced JAK/STAT signaling (and *p*-STAT3) by the JAK inhibitor ruxolitinib prevented IL6 transcription and secretion (Figure 6C,D). This suggests that a co-operative, second IL1α-independent signal is required for full activation of the IL6 gene locus. We did not see similar inhibition of inducible IL8 in HBF (data not shown).

### 3.7. Src-Dependent Signaling Is Required for IL8 Expression by HBF in Response to Eosinophil-Derived Supernatants, While NFκB Is Dispensable

In addition to releasing IL6, HBF activated by soluble mediators from IL3IgG-stimulated eosinophils also secrete high levels of pro-neutrophilic and pro-inflammatory cytokine IL8. However, when we tried to inhibit eosinophil supernatant-induced IL8 in HBF via the blockade of NFκB signaling with BMS-345541, there was no effect (Figure 7A). This suggested that IL1α uses an alternative signaling pathway to induce IL8 gene expression. Considering that proto-oncogene tyrosine-protein kinase Src has previously been reported to signal downstream of IL1β to lead to IL8 expression [28], we hypothesized that this kinase plays a role in the secretion of IL8 by eosinophil soluble mediator activated-HBF. Our data demonstrate that treatment with PP2, selective inhibitor of Src-family kinases, disrupts the ability of activated HBF to release IL8, when compared to the inactive, analogous small molecule, PP3 (Figure 7B) and that this effect is transcriptionally regulated (Figure 7C). Interestingly, treatment with PP2 did not disrupt IL6 secretion under the same conditions (Figure 7D). Our data demonstrate that HBF stimulated by soluble mediators from IL3IgG-activated eosinophils release IL8 through the Src-family kinases-mediated signaling, in contrast to the NFκB and JAK/STAT signaling required for IL6.

## 4. Discussion

In this study, our primary observation is that the blockade of IL1α is sufficient to prevent the induction of IL6 and IL8 expression in HBF by eosinophil-derived soluble mediators and is associated with eosinophilic inflammation in human airways after antigen challenge. This finding suggests that IL1α is a key component of the milieu of soluble mediators released upon eosinophil degranulation and cytolysis that drives pro-inflammatory signaling in bronchial fibroblasts.

Interleukin-1 family cytokines play an important role in both the innate and adaptive immune response and are implicated in eosinophilic inflammation and airway remodeling [25,29]. The IL1 receptor is bound by both and IL1α and IL1β, with similar downstream signaling responses. IL1α is expressed ubiquitously in many immune and structural cells [30] and acts as an alarmin. Activated eosinophils have also been found to express IL1α [31]. In contrast, IL1β has a more limited expression, mainly in immune cells, including our findings that it is expressed in stimulated eosinophils [25,26]. Thus, it is somewhat surprising that we have observed that IL1α, rather than IL1β, appears to be the key ligand in the activation of bronchial fibroblasts incubated with eosinophil-derived soluble mediators. However, IL1α has a role in the induction of sterile inflammation from necrotic cells, including neutrophils [32]. Thus, it is intriguing to speculate that activated eosinophils undergoing cytolysis may also be a source of IL1α, which then can signal to other cellular compartments in the allergically-inflamed airway. For example, it is known that IL1α co-localizes with nuclear material, and extracellular nuclear contents, including DNA, have been found in association with eosinophilic inflammation [33,34]. Although IL1α can induce neutrophil recruitment in other contexts, we previously found that eosinophil-derived supernatants did not induce neutrophil chemotaxis [6]. Rather, in this context, it appears that eosinophil-derived IL1α may only play a role in neutrophil chemotaxis via the induction of bronchial fibroblast-derived IL8, which we have found in sufficient quantities to recruit neutrophils [6]. Transduction of immune signals via the mesenchyme is increasingly recognized as an important component of a variety of inflammatory disorders and contributes to tissue remodeling [35,36]. In other contexts, IL1α has been shown to signal to mesenchymal cells with subsequent release of pro-neutrophilic signals [37].

To determine the potential in vivo relevance of this finding, we utilized segmental antigen bronchoprovocation, a powerful tool to interrogate the mechanism of allergic inflammation in the in vivo context [38]. Using BAL samples obtained from Day 0 (just prior to antigen challenge) versus Day 2 (two days after antigen challenge), a time characterized by intense eosinophilic inflammation, we found a compelling correlation between IL1α levels and eosinophilic inflammation. This finding supports the plausibility of IL1α as a component of the inflammatory milieu of the allergic airway. While the cellular source of IL1α cannot be discerned from these experiments, combined with our ex vivo investigations, it seems that, in addition to other known sources of this cytokine [30], IL1α may originate in part from eosinophilic cytolysis and degranulation. However, given that expression or release of IL1α in an autocrine fashion can be induced by IL1 family cytokines or other agonists [39,40,41], we cannot completely exclude that the source of IL1α is from HBF. However, this possibility appears unlikely due to the ability of IL1α neutralizing antibody to inhibit eosinophil soluble factor-mediated NFκB signaling within 30 min of activation.

A second key conclusion of this study is that IL1α signals via the IL1R and NFκB to induce IL6 expression and secretion in HBF, but uses alternative, NFκB-independent signaling to induce IL8. IL1 family cytokines signal through the IL1 receptor (IL1R), a member of the IL1/Toll-like receptor superfamily. IL1R has a Toll-IL1R (TIR) domain, which couples the receptor to MyD88 and IL1 receptor-associated kinase (IRAK), mediating downstream activation of IκB kinase (Ikk), phosphorylation of IκB (followed by ubiquitination/degradation) and activation of NFκB. We have found that there is activation of NFκB signaling by eosinophil supernatants, consistent with activation of IL1R. Furthermore, blockade of IL1α inhibited this activation, confirming the role of IL1α in its activation. Finally, IL6 expression in HBF stimulated with eosinophil supernatants is dependent on NFκB activation based on the use of a small molecule inhibitor of Ikk. Interestingly, downstream signaling to IL6 and IL8 activation seems to diverge in this model, with canonical NFκB signaling only required for IL6 activation. For example, we additionally found that Src was required for the expression of IL8 in HBF in response to eosinophil soluble mediators, whereas NFκB was dispensable for its activation. IL1 cytokines, in particular IL1β, have been reported to activate Src signaling in other contexts and cell types [28]. This suggests that it is possible that IL1α uses signaling via Src to induce IL8 expression and secretion, although we cannot exclude activation via an alternative agonist in the eosinophil-derived supernatants. Finally, we also observed dependency of IL6 on JAK/STAT signaling, indicating that co-operative signaling from other non-IL1α ligands is at play in the induction of IL6.

A potential strength of our approach is the use of well-characterized eosinophil cultures from humans. The fidelity of the IL3-induced phenotype to airway eosinophils is well established, making them a highly relevant tool to understand the behavior and influence of eosinophils in airway inflammation and remodeling. Secondly, we have coupled these cells with primary cultures of bronchial fibroblasts. While limitations of primary fibroblast culture are understood, we feel that this is a relevant model to study signaling interactions and enable mechanistic investigations. We acknowledge that these studies are performed using an ex vivo system that may not recapitulate the findings in vivo. However, to address this limitation we did examine the correlation between IL1α and eosinophilic inflammation in the segmental antigen challenge model of human allergic airway inflammation.

Follow-on studies to isolate BAL eosinophils in the allergically inflamed airway, followed by induction of cytolysis and examination of the IL1α dependency of the effect could also further support its role in in vivo allergic inflammation. Additional studies are needed to further define the mechanism through which IL1R-mediated signaling controls the release of IL8 and the mode of co-operation between NFκB and JAK/STAT signaling leading to the release of IL6 in fibroblasts activated by eosinophil soluble mediators.

## 5. Conclusions

In conclusion, we have found that IL1α appears to be a critical component of the soluble eosinophil-derived milieu that drives pro-inflammatory bronchial fibroblast responses and is associated with eosinophilic inflammation in the human airway following SBP-Ag. Further, we identified that IL1α-dependent NFκB signaling works in tandem with JAK/STAT and Src to induce the release of IL6 and IL8 from HBF. Taken together, these data suggest an important role for IL1α in mediating paracrine signals from eosinophils undergoing cytolysis in the allergic asthmatic airway. Disrupting these downstream signaling intermediates could be a strategy to mitigate the airway remodeling effects of eosinophilic inflammation.

## Figures and Tables

**Figure 1 cells-10-00528-f001:**
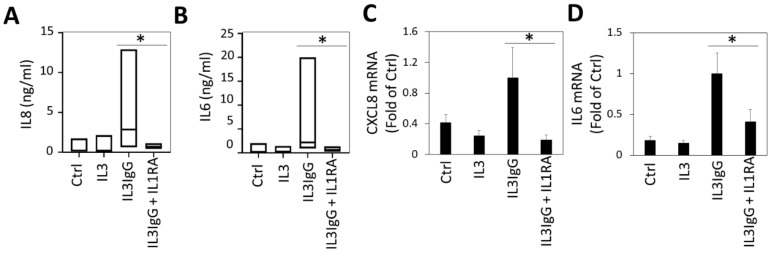
Interleukin 6 (IL6) and IL8 expression and secretion by human bronchial fibroblasts stimulated with eosinophil-derived soluble mediators requires signaling via the IL1 receptor. Human bronchial fibroblasts (HBF) were incubated with IL1 receptor antagonist (IL1RA, 100 ng/mL) or vehicle (0.1% bovine serum albumin (BSA) in PBS) for 30 min, and subsequently stimulated with eosinophil supernatants (IL3 or IL3IgG) or basal medium (Ctrl). Twenty-four hours later, HBF supernatants were analyzed via ELISA for levels of IL8 (**A**) and IL6 (**B**) (*n* = 8 for all conditions), and paired Student’s *t*-test was used to test for statistical significance (* *p* < 0.05). HBF lysates were analyzed for mRNA levels of CXCL8 (**C**) and IL6 (**D**) via RT-qPCR (*n* = 3 for all conditions) and analyzed by setting IL3IgG as a reference (mean ± sd) and using unpaired Student’s *t*-test to test for statistical significance (* *p* < 0.05).

**Figure 2 cells-10-00528-f002:**
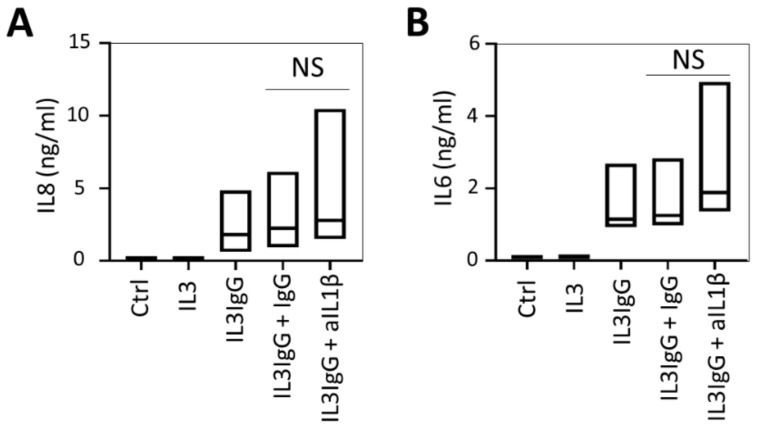
Release of IL6 and IL8 by human bronchial fibroblasts stimulated with products from activated eosinophils is independent of IL1β. HBF were stimulated with eosinophil supernatants (IL3 or IL3IgG) or basal medium (Ctrl). IL3IgG eosinophil supernatants were incubated for 1 h with IgG control antibody (1 µg/mL) or IL1β neutralizing antibody (aIL1β, 1 µg/mL) prior to stimulation, as indicated. Twenty-four hours later, HBF supernatants were tested for IL8 (**A**) and IL6 (**B**) via ELISA (*n* = 4 for all conditions). Paired Student’s *t*-test was used to test for statistically significant differences (NS = not significant).

**Figure 3 cells-10-00528-f003:**
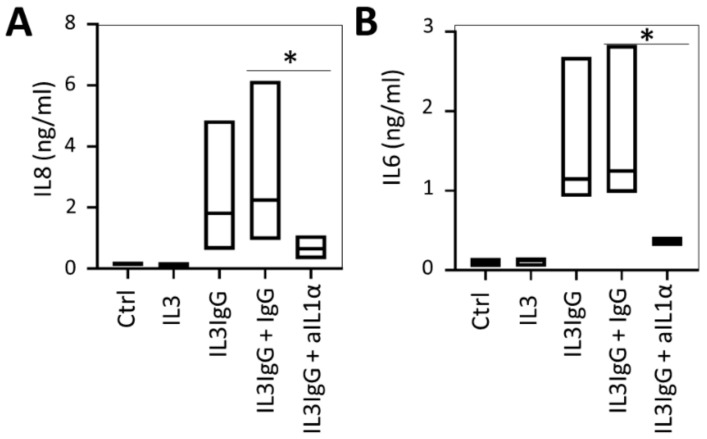
Release of IL6 and IL8 by human bronchial fibroblasts stimulated with eosinophil soluble mediators is dependent on activation by IL1α. HBF were stimulated with eosinophil supernatants (IL3 or IL3IgG) or basal medium (Ctrl). IL3IgG supernatants were incubated for 1 h with either IgG control antibody (1 µg/mL) or IL1α neutralizing antibody (aIL1α, 1 µg/mL) prior to stimulation, as indicated. Twenty-four hours later, HBF supernatants were collected, and IL8 (**A**) and IL6 (**B**) levels were analyzed via ELISA (*n* = 4 for all conditions). Paired Student’s *t*-test was used to test for statistically significant differences (* *p* < 0.05).

**Figure 4 cells-10-00528-f004:**
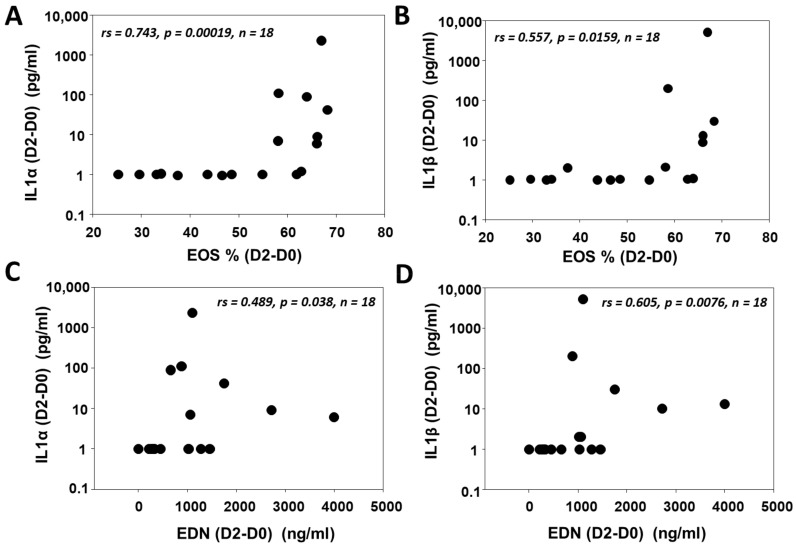
Eosinophilic inflammation is associated with IL1α and IL1β in bronchoalveolar lavage (BAL) after a segmental allergen challenge in allergic patients with mild asthma. ELISA against IL1α (**A**,**C**), IL1β (**B**,**D**) and eosinophil-derived neurotoxin (EDN) (**C**,**D**), as well as the quantification of purified eosinophils (**A**,**B**), was performed from BAL samples before segmental allergen challenge (SBP-Ag) (D0) and 48 h after the challenge (D2) (paired values from *n* = 18 subjects). Changes from D0 to D2 are presented for all variables. Spearman correlation was utilized for statistical analysis.

**Figure 5 cells-10-00528-f005:**
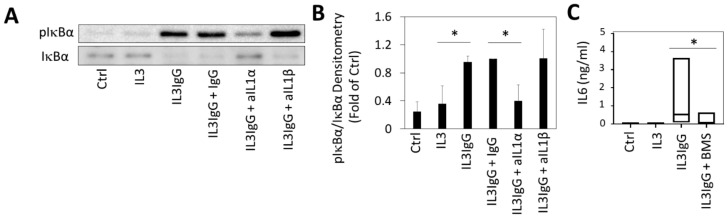
Eosinophil lysis products induce the production of IL6 by human bronchial fibroblasts through the IL1α-dependent nuclear factor kappa-light-chain-enhancer of activated B cell (NFκB) signaling mechanism. (**A**) IL3IgG supernatants were incubated for 1 h with either IgG control antibody (1 µg/mL), IL1α neutralizing antibody (aIL1α, 1 µg/mL) or IL1β neutralizing antibody (aIL1β, 1 µg/mL) prior to stimulation of HBF. HBF were alternatively incubated with basal medium (Ctrl), IL3 or IL3IgG eosinophil supernatants as a control. Thirty minutes after stimulation, HBF protein lysates were obtained and subjected to SDS-PAGE and Western blotting with indicated antibodies. (**B**) Densitometry of Western blots from A. (*n* = 3 for all conditions) was done in ImageJ (mean ± sd) and analyzed by unpaired Student’s *t*-test with Bonferroni correction to determine statistically significant differences (* *p* < 0.025). (**C**) HBF were treated with IκB kinase inhibitor (10 µM BMS-345541 (BMS)) or vehicle (dimethyl sulfoxide (DMSO)) prior to stimulation with eosinophil supernatants for 24 h. HBF supernatants were collected (*n* = 7 for all conditions) and levels of IL6 were determined via ELISA. Paired Student’s *t*-test was used to test for statistically significant differences (* *p* < 0.05).

**Figure 6 cells-10-00528-f006:**
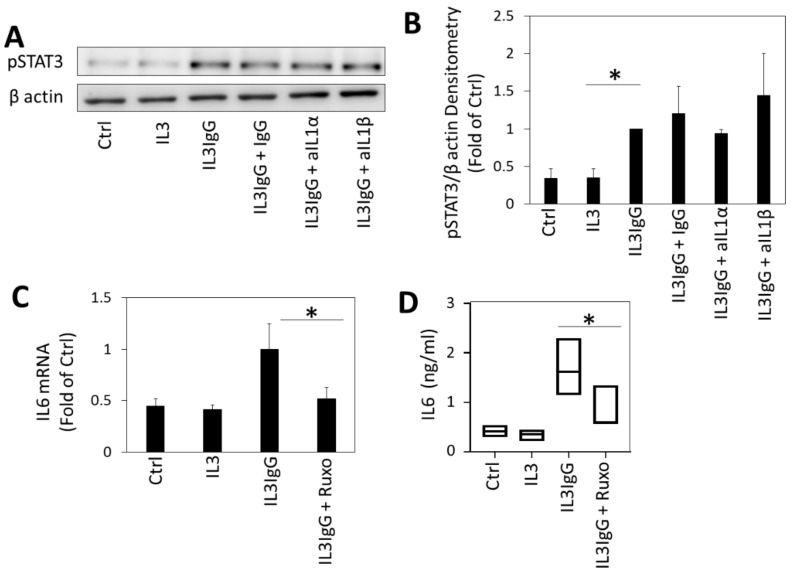
Eosinophil-derived supernatants lead to the production of IL6 by HBF through the IL1-independent Janus Kinase (JAK)/Signal Transducer and Activator of Transcription Protein (STAT) signaling mechanism. (**A**) IL3IgG supernatants were incubated for 1 h with either IgG control antibody (1 µg/mL), IL1α neutralizing antibody (aIL1α, 1 µg/mL) or IL1β neutralizing antibody (aIL1β, 1 µg/mL) prior to stimulation of HBF. HBF were alternatively incubated with basal medium (Ctrl), IL3 or IL3IgG eosinophil supernatants as a control. Thirty minutes after stimulation, HBF protein lysates were obtained and subjected to SDS-PAGE and Western blotting with indicated antibodies. (**B**) Densitometry of Western blots from A. (*n* = 3 for all conditions) was done in ImageJ (mean ± sd). C-D. HBF were treated with Janus-associated kinase (JAK) inhibitor (100 nM ruxolitinib (Ruxo)) or vehicle (ethanol) prior to stimulation with eosinophil supernatants for 24 h. mRNA levels of IL6 in HBF (*n* = 3 for all conditions) were assessed via RT-qPCR (**C**). Levels of IL6 in HBF supernatant (*n* = 3 for all conditions) were assessed via ELISA (**D**). Unpaired (**B**,**C**) or paired (**D**) Student’s *t*-test was used to test for statistically significant differences (* *p* < 0.05).

**Figure 7 cells-10-00528-f007:**
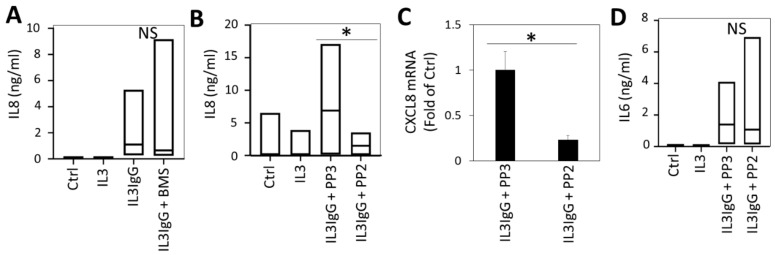
Eosinophil lysis products lead to the release of IL8 by human bronchial fibroblasts through the Src family kinase signaling mechanism. HBF were treated with IκB kinase inhibitor (10 µM BMS-345541 (BMS)), or vehicle (DMSO) (**A**), or inhibitor of Src-family kinases (10 µM PP2 (or PP3 control)) (**B**–**D**) prior to stimulation with eosinophil supernatants. Twenty-four hours later, HBF lysates were assessed for levels of IL8 or IL6 (*n* = 7 for all conditions in A; *n* = 3 for Ctrl and IL3 conditions and *n* = 4 for IL3IgG + PP3 and IL3IgG + PP2 conditions in (**B**,**D**)) via ELISA (**A**–**D**). At the same time point, mRNA levels of CXCL8 in HBF (*n* = 3 for all conditions) were assessed via RT-qPCR (mean ± sd). Paired (**A**–**D**) or unpaired (**C**) Student’s *t*-test was used to test for statistically significant differences (* *p* < 0.05).

## Data Availability

No data included in this manuscript have been placed in a public repository. Data can be made available upon request to the corresponding author.

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
