# Peer review of "Interleukin-1α Is a Critical Mediator of the Response of Human Bronchial Fibroblasts to Eosinophilic Inflammation"

_cells, 2021, doi:10.3390/cells10030528_

Round 1

Reviewer 1 Report

In this study, the authors report that eosinophil soluble mediators stimulate human bronchial fibroblasts (HBF) through IL1R by IL1α but not IL1β. Neutralization of IL1α attenuates the production of IL8 and IL6 by HBF stimulated with the soluble mediators released from IL3/IgG-activated eosinophils. The authors highlight that the release of IL8 by HBF is through Src-family kinases-mediated signaling, and that of IL6 is dependent on NFkB and JAK/STAT signaling. The data suggest that activated eosinophil contributes to airway inflammation, at least in part, through IL1α-mediated signaling in HBF.

General comments. The data are convincing, well presented, and discussed. I have only a few minor suggestions.

Minor comments and corrections:

The authors can provide shorter figure legends. It is unnecessary to give details on the production of eosinophil supernatant. There are sufficient details in the "Materials and Methods" section.

Line 100: collagenase

Section 2.3: What are the detection limits for IL1α, IL1β, and EDN ELISA, respectively?

Lines 273 & 275 : 1 µg/ml of IL1α

Line 527: The remaining authors declare no conflict of interest.

Appendix D: Table 18 I

Author Response

The authors can provide shorter figure legends. It is unnecessary to give details on the production of eosinophil supernatant. There are sufficient details in the "Materials and Methods" section.

We appreciate the reviewer’s comment and have now edited our figure legends for brevity and clarity. To see the modifications, please view the tracked changes in the updated manuscript document.

Line 100: collagenase

Thank you for detecting this spelling error. It has now been fixed.

Section 2.3: What are the detection limits for IL1α, IL1β, and EDN ELISA, respectively?

The detection limits of the IL1α, IL1β and EDN ELISA kits have now been addressed at the end of Section 2.3 by stating “The IL1α and IL1β DuoSet Development ELISA kits, and EDN ELISA kit had sensitivities of 1 pg/ml and 1 ng/ml, respectively.”

Lines 273 & 275 : 1 µg/ml of IL1α

Thank you for detecting this error. It has now been fixed in both locations.

Line 527: The remaining authors declare no conflict of interest.

We have now corrected this grammatical error.

Appendix D: Table 18 I

Thank you for pointing this out. We have corrected the erroneous numbering.

Reviewer 2 Report

This paper describes IL1α as a crucial component of the soluble eosinophil-derived network that guides pro- inflammatory bronchial fibroblast responses. Its main strength is model – cells taken from patients and not cell lines.

Authors have explained the concept of their study and described materials and methods in details. However, it is no clear how many subjects were in control group and how many in test group(s) that could impact statistics.

The other remark concerns figure legends: there is no need for such thorough explanation which is essentially explanation of methods that should just be a part of the text. Legends are supposed to be self-explanatory, but repeating whole procedure is a bit too much and it makes reading more, instead of less, complex.

Author Response

Authors have explained the concept of their study and described materials and methods in details. However, it is no clear how many subjects were in control group and how many in test group(s) that could impact statistics.

We thank the reviewer for this comment. We have clarified this for both in vivo and ex vivo studies.

For in vivo studies, in Section 2.1 (last paragraph), we added “In all subjects, BAL was performed…” and in the legend for Figure 4, we added “(paired values from N=18 subjects).” In this way, we clarified that all 18 subjects underwent segmental broncho provocation with an allergen (SBP-Ag) and the comparison was made between samples collected pre and post challenge in each subject.

For ex vivo studies, the number of experimental repeats were included in figure legends. As noted, the number of experimental repeats were equivalent between conditions within each experiment. The only exception to this was found in Figure 7 B and D where Ctrl and IL3 conditions were repeated 3 times whereas IL3IgG + PP3 and IL3IgG + PP2 were repeated 4 times.    

The other remark concerns figure legends: there is no need for such thorough explanation which is essentially explanation of methods that should just be a part of the text. Legends are supposed to be self-explanatory, but repeating whole procedure is a bit too much and it makes reading more, instead of less, complex.

We appreciate the reviewer’s comment and have now edited our figure legends for brevity and clarity. To see the modifications, please view the tracked changes in the updated manuscript document.